# Six-Dimensional Manifold with Symmetric Signature in a Unified Theory of Gravity and Electromagnetism

**Nikolay Popov * and Ivan Matveev ***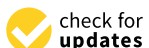

Federal Research Center "Computer Science and Control" of the Russian Academy of Sciences,
Vavilov Str., 44/2, 119333 Moscow, Russia
* Correspondence: nnpopov@mail.ru (N.P.); matveev@frccsc.ru (I.M.)

**Abstract:** A six dimensional manifold of symmetric signature $(3,3)$ is proposed as a space structure for building combined theory of gravity and electromagnetism. Special metric tensor is proposed, yielding the space which combines the properties of Riemann, Weyl and Finsler spaces. Geodesic line equations are constructed where coefficients can be divided into depending on the metric tensor (relating to the gravitational interaction) and depending on the vector field (relating to the electromagnetic interaction). If there is no gravity, the geodesics turn into the equations of charge motion in the electromagnetic field. Furthermore, symmetric six-dimensional electrodynamics can be reduced to traditional four-dimensional Maxwell system, where two additional time dimensions are compactified. A purely geometrical interpretation of the concept of electromagnetic field and point electric charge is proposed.

**Keywords:** Riemann–Weyl–Finsler spaces; space-time structure; unified field theory

## 1. Introduction

The question of combining the theories of gravitation and electromagnetism has interested many researchers. Approaches to unification of these theories were proposed by Einstein [1], Eddington [2], Weyl [3], Cartan [4] and others. These works use the four-dimensional signature manifold $(1,3)$, which is the space we observe in ordinary life, with one temporal and three spatial coordinates. However, these unifying theories have fundamental shortcomings that forced researchers to introduce additional dimensions. So the five-dimensional Kaluza–Klein model combining gravitation and electromagnetism [5,6] appeared, as well as its various generalizations, such as supersymmetric models [7,8], supergravity theory proposed by Friedman, Nieuwenhuizen and Ferrara [9], Deser and Zumino [10], and others [11].

In these models, some compact manifold $B$ is added to the four-dimensional space-time manifold $M$ as a component of the direct product. The resulting manifold $M \otimes B$ represents an extended space on the basis of which unified theories of gauge fields are constructed. The gauge fields are induced by symmetry groups $B$. It should be noted that the number of additional dimensions introduced by the $B$ manifold can be quite large. For example, the minimum number of $B$ dimensions required to construct a gauge theory of superunification based on the structural group $SU(3) \otimes SU(2) \otimes U(1)$ is 7 [12].

The relation to the geometrical nature of the complementary dimensions of superspace is ambiguous. The simplest way to formulate a distinction between the basic dimensions of manifold $M$ and the additional dimensions $B$ is in terms of the bundle theory [13]. If the manifold $M \otimes B$ is identified with the bundle, $M$ is regarded as its base and $B$ as a typical layer, then the pair $(M, B)$ defines a trivial vector bundle over the base $M$. The typical layer $B$ may not be directly related to the geometric structure of the base $M$, so the additional dimensions of the layer $B$ do not necessarily have a direct geometric interpretation associated with the geometric nature of the basic dimensions. This greatly

complicates the task of creating a unified and purely geometric field theory of interactions. To the best of our knowledge no sufficiently convincing geometrical constructions have been proposed so far.

In this paper an attempt is made to develop the foundations of a unified theory of gravitation and electromagnetism based on the space-time with symmetric signature (i.e., equal number of spatial and temporal dimensions), which extends usual space of signature $(1, 3)$. The concept can be summarized as follows:

1. unified geometrical theory of physical fields of matter is a theory of bundles, in which the base of the bundle is a space-time manifold;
2. structural groups of the bundle induce gauge fields;
3. geometrical structure of the bundle base is chosen so that the structural groups are the symmetry groups of the base;
4. gauge fields are connections of structural groups functionally related to real physical fields.

The smallest extension of the usual space to symmetric signature is obtained by adding two temporal dimensions, resulting in the signature of $(3, 3)$. It turns out that such space is sufficient for constructing the unified theory according to statements above.

Structure of the work is as follows. A space combining elements of the structures of Riemann, Weyl and Finsler spaces is introduced for a uniform description of gravitational and electromagnetic interactions. A definition of the geodesic in this space is given, and it is shown that when moving along the geodesic, the space remains homogeneous and isotropic. The system of geodesic equations taking into account the presence of gravitational and electromagnetic fields is derived. It is shown that in the framework of the considered formalism the interaction of an electromagnetic field with a current ($A - J$ interaction) is possible only in spaces of dimension greater than four. If a gravitational field is absent, the geodesic equations take the form of the Lorentz equation describing the motion of a unit charge in the electromagnetic field. On this basis, the equations of electrodynamics in the pseudo-Euclidean signature space $(3, 3)$ are constructed and the notions of charge and current densities, which have a purely geometric nature, are introduced. The connection between the six-dimensional and traditional systems of Maxwell's equations is outlined. It is established that the class of admissible currents is determined by a group of local eigenmovements of the metric of the introduced space.

## 2. Basic Definitions

The basis of the proposed geometric model of gravitational and electromagnetic fields is the notion of a metric dynamic six-dimensional space-time of signature $(3, 3)$. Here "dynamic" means the metric tensor depends on the local velocity. Such a space partially exhibits the properties of Weyl [3] and Finsler [14,15] spaces.

The Weyl space is defined by a family of conformally equivalent metrics $\lambda(x)$, $g(x)$, where $\lambda(x)$ is an arbitrary positive function, and $g(x)$ is the field of the metric tensor on the manifold. This type of space was used by Weyl to construct a unified theory of gravitational and electromagnetic fields. The main drawback of this theory is the requirement of inhomogeneity of physical space, i.e., the absence of a unified scale at different points in space, which is not confirmed by observations available at the moment.

Finsler geometry is the geometry of metric spaces with internal local anisotropy [15,16]. The Finsler metric tensor depends not only on the points of the basic manifold, as it is the case in Riemannian space, but also on the values of local velocities at these points, i.e., it has the form $g_{ij}(x, \dot{x})$. Accordingly, the physical fields in Finsler space, in addition to space-time coordinates, turn out to depend on local velocities. The disadvantage of using Finsler space for constructing a unified field theory is the requirement of local anisotropy. Note that to date there are no convincing indications of local anisotropy of physical space-time. Moreover, the use of Finsler geometry in field theories of Kaluza–Klein type [5] is characterized by a large variety of possible structures and by the resulting problem of identification of new elements of structure with physical observables.

### 3. Rwf-Spaces

The spaces combining properties of Riemann, Weyl and Finsler spaces, where a unified theory of gravitational and electromagnetic interactions can be described in a unified way, are introduced as follows. Let $M^n$ be an $n$-dimensional connected manifold and $TM^n$ be a tangent bundle over the base of $M^n$. Let $g_{ij}(x, y)$, $(x, y) \in U \otimes T_x M^n$, where $T_x M^n$ is a tangent layer over the point $x \in U$. If the relation holds

$$g_{ij}(x, y) = \lambda(x, y) g_{ij}(x) , \tag{1}$$

where $\lambda(x, y)$ is a positive function of two variables, then the tensor $g_{ij}(x, y)$ will be called a generalized metric tensor at a given point $x$ for a given vector $y$. Thus, the generalized metric is not only a function of a point in space, but also of the direction vector given at that point.

**Definition 1.** *Riemann–Weil–Finsler space (RWF-space) is the manifold $M^n$ in which the metric tensor field of the form* (1) *twice continuously differentiable by the arguments x, y, twice covariant, symmetric and nondegenerate is defined.*

A differential bilinear form can be introduced in RWF-space

$$ds^2 = g_{ij}(x, \dot{x}) dx^i dx^j , \tag{2}$$

where $\dot{x}^i = dx^i / ds$, $x^i(s)$ is an arbitrary continuously differentiable line in some local region of space.

In spaces with affine connectivity, a line $x^i(s)$ is *geodesic* if its velocity vector is parallel along itself, i.e., the condition $\nabla_{\dot{x}} \dot{x} = 0$. In coordinate form

$$\nabla_{\dot{x}} (\dot{x})^j = \ddot{x}^j + \Gamma^j_{ki} \dot{x}^k \dot{x}^i = 0 , \tag{3}$$

where $\nabla_{\dot{x}} (\dot{x})^j$ is the covariant derivative along the vector $\dot{x}$, $\Gamma^j_{ki}$ is the connectivity, which depends on two arguments $x$ and $\dot{x}$ in general case. In order to ensure homogeneity and isotropy of space when moving along geodesics, one restriction should be added to the definition of geodesics.

**Definition 2.** *The geodesic $x^i(s)$ will be called geodesic in RWF-space (RWF-geodesic) if the condition takes place at each point of the line*

$$\lambda(x(s), \dot{x}(s)) = 1 , \tag{4}$$

*where $\lambda(x, y)$ is some positive continuous function for both arguments.*

From the definition of RWF-geodesic it follows that on any such curve $x(s)$ the relation holds

$$g_{ij}(x(s), \dot{x}(s)) = g_{ij}(x(s)) . \tag{5}$$

This means that scale remains unchanged and there is no local anisotropy of space for parallel movement along the RWF-geodesic.

Let us now turn to the question of the specific form of representation of the function $\lambda(x, \dot{x})$. Let $\lambda(x(s), \dot{x}(s))$ be an arbitrary function continuous in both arguments in some neighborhood $U$ of point $x(s_0)$, where $x(s)$ is an arbitrary continuously differentiable function in the this neighborhood.

**Lemma 1.** *It is always possible to find a continuous vector field $A_k(x)$ in the tangent bundle $TU$ over the base $U$ such that*

$$\lambda(x(s), \dot{x}(s)) = \exp\left( A_k(x(s), \dot{x}(s)) \dot{x}^k(s) \right) . \tag{6}$$

**Proof.** Let the local coordinate system in the neighborhood of the point $x(s_0) \in U$ be chosen such that at this point the vector $\dot{x}(s_0) \in T_{x(s_0)}U$ has only one non-zero component, for example, $\dot{x}^1(s_0)$. Then put, by definition

$$A_1(x(s_0), \dot{x}(s_0)) = \frac{\ln \lambda(x(s_0), \dot{x}(s_0))}{\dot{x}^1(s_0)} . \tag{7}$$

The remaining components of the vector $A_i(x, \dot{x})$ at the point $x(s_0)$ are chosen arbitrarily. Then at the point $x(s_0)$ the relation holds

$$\lambda(x(s_0), \dot{x}(s_0)) = \exp\left(A_k(x(s_0), \dot{x}(s_0))\dot{x}^k(s_0)\right) . \tag{8}$$

Because of the continuity of $\lambda(x, \dot{x})$ with respect to both arguments and the continuity of $\dot{x}^k(s_0)$, the relation (8) can be extended in a continuous way to some neighborhood of $x(s_0)$. $\square$

Further we restrict ourselves to consider a particular case of $A_k$ independent of $\dot{x}$.

As follows from the construction, the choice of the vector field $A(x)$ turns out to be ambiguous. If the line $x(s)$ is RWF-geodesic, then according to (6) the vector field $A(x(s))$ is orthogonal to the velocity vector field $\dot{x}(s)$ at each point of $x(s)$.

### 4. Equations of RWF-Geodesics

To find the RWF-geodesic we will use the Lagrangian formalism. We choose the bilinear quadratic form $L = g_{ij}(x, \dot{x})\dot{x}^i\dot{x}^j$ as the Lagrangian $L$. According to the formula (2), $L = 1$. By virtue of the relation (6)

$$g_{ij}(x, \dot{x}) = \exp(A_k\dot{x}^k)g_{ij}(x) , \tag{9}$$

and according to (4) and (6) on the geodesic line $x(s)$ the following holds

$$A_k(x(s))\dot{x}^k(s) = 0 . \tag{10}$$

The geodesic equations are given by the system of Euler-Lagrange differential equations

$$\frac{\partial L}{\partial x^k} - \frac{d}{ds}\frac{\partial L}{\partial \dot{x}^k} = 0 , \quad k = \overline{1, n} . \tag{11}$$

By virtue of the condition (10) we have

$$\frac{\partial L}{\partial x^k} = \frac{1}{2}g_{ij,k}(x)\dot{x}^i\dot{x}^j + A_{l,k}(x)\dot{x}^l ,$$
$$\frac{d}{ds}\frac{\partial L}{\partial \dot{x}^k} = g_{ij,k}(x)\dot{x}^j\dot{x}^j + g_{ik}\ddot{x}^i + A_{k,l}(x)\dot{x}^l , \tag{12}$$

where $g_{ij,k} = \partial g_{ij}/\partial x^k$, $A_{k,l} = \partial A_k/\partial x^l$. Substituting these relations into the Euler-Lagrange equations, we finally obtain a system of second-order differential equations for geodesics

$$\ddot{x}^p + \frac{1}{2}g^{pk}(g_{ik,j} + g_{jk,i} - g_{ij,k})\dot{x}^i\dot{x}^j + (A_{k,l} - A_{l,k})g^{pk}\dot{x}^l g_{ij}\dot{x}^i\dot{x}^j = 0 , \quad p = \overline{1, n} . \tag{13}$$

Introducing standard notations

$$\Gamma_{ij}^p(x) = \frac{1}{2}g^{pk}(x)\left(g_{ik,j}(x) + g_{jk,i}(x) - g_{ij,k}(x)\right) , \quad F_{ij}(x) = A_{j,i}(x) - A_{i,j}(x) , \tag{14}$$

equations of geodesic (13) in RWF-space can be represented in the form

$$\ddot{x}^p + \Gamma_{ij}^p\dot{x}^i\dot{x}^j + F_{lk}\dot{x}^l g^{pk}g_{ij}\dot{x}^i\dot{x}^j = 0 , \tag{15}$$

or in the form

$$\ddot{x}^p + \Gamma_{ij}^p \dot{x}^i \dot{x}^j + g^{pk} g_{lj} \dot{x}^l F_{ki} \dot{x}^i \dot{x}^j = 0 , \tag{16}$$

The second term in (15) and (16) at $\Gamma_{ij}^p$ depends only on $x$ but not on $\dot{x}$ and corresponds to Riemann geometry. These two representations lead to different geometric structures. By introducing a generalized connectivity symmetric by indices $i, j$

$$\hat{\Gamma}_{ij}^p = \Gamma_{ij}^p + F_{lk} \dot{x}^l g^{pk} g_{ij} \tag{17}$$

and with anti-symmetric connectivity

$$\hat{\hat{\Gamma}}_{ij}^p = \Gamma_{ij}^p + g^{pk} g_{lj} \dot{x}^l F_{ik} , \tag{18}$$

Equations (15) and (16) can be represented in short form

$$\ddot{x}^p + \hat{\Gamma}_{ij}^p \dot{x}^i \dot{x}^j = 0 , \tag{19}$$

$$\ddot{x}^p + \hat{\hat{\Gamma}}_{ij}^p \dot{x}^i \dot{x}^j = 0 , \tag{20}$$

where the first term in the right-hand sides of (17) and (18) is responsible for gravitation, and the second is for electromagnetism.

Further in this paper we treat electromagnetic aspect of (19) and (20). In Section 6 we assume zero gravitation to ease calculations and than reduce to Maxwell equations, which are derived without gravity considerations at all. We plan to present gravitation aspect in the next article.

## 5. Equations of Electrodynamics in Six-Dimensional RWF-Space

Using the vector field $A_k(x)$ included in the definition of RWF-metrics, we derive the basic equations of six-dimensional electrodynamics and introduce the concepts of charge density and current, which are purely geometric in nature.

From the real vector field $A_k(x)$ included in the definition of RWF-metrics, one can always construct the field of the bivalent antisymmetric tensor $F_{ij} = A_{j,i} - A_{i,j}$, $i, j = \overline{1,6}$, representing the rotor of the vector field $A_k$. The operation of taking the gradient of this tensor gives the identical zero by virtue of the Bianchi identity

$$F_{ij,k} + F_{ki,j} + F_{jk,i} \equiv 0 , \quad i, j, k = \overline{1,6} . \tag{21}$$

The identity (21) is valid for spaces of arbitrary dimension and signature. It is not related to the type of the space metric and remains covariant with respect to any nondegenerate coordinate transformations. Note also that the vector field $A_k$ giving rise to this identity can be chosen arbitrarily. Another covariant relation, which can be constructed using the antisymmetric tensor $F^{ij}$, has the form

$$D_i F^{ij} = 0 , \quad j = \overline{1,6} , \tag{22}$$

where $D_i$ is the covariant derivative of the parameter $x^i$. Clearly, if the identity (22) holds in any coordinate system, it holds in any other system as well. However, unlike the identity (21), the system of equations (22) depends on the type of space-time metric. Let us partition the system of equations (22) into two subsystems

$$D_1 F^{1j} + \cdots + D_4 F^{4j} = -D_5 F^{5j} - D_6 F^{6j} , \quad j = \overline{1,4} , \tag{23}$$

$$\begin{aligned} D_1 F^{15} + \cdots + D_4 F^{45} &= -D_6 F^{65} , \\ D_1 F^{16} + \cdots + D_4 F^{46} &= -D_5 F^{56} . \end{aligned} \tag{24}$$

The system (23) remains covariant with respect to any coordinate transformations from the group $GL(4, \mathbb{R})$ while the system (24) is covariant with respect to transformations from the group $GL(2, \mathbb{R})$. Let us introduce the notation

$$\frac{J^j}{c} = -D_5 F^{5j} - D_6 F^{6j} \ , \ \ j = \overline{1, 4} \ . \tag{25}$$

From the definition it follows that this object is a four-component contravariant vector field in a four-dimensional submanifold of six-dimensional RWF-space.

**Definition 3.** *A four-dimensional current density vector is the vector field $J^j(x)$, $j = \overline{1, 4}$ in the four-dimensional submanifold of six-dimensional RWF-space.*

**Definition 4.** *Charge density is the value $\rho(x) = J^4(x)/c$.*

These definitions are a tribute to the established tradition, since current density and charge density are closely related to the well-known phenomenological concepts of electric current density and electric charge density. These very notions are used further, although it would be more consistent to operate only with the components of the six-dimensional electromagnetic tensor $F^{ij}$.

So, the relation (23), according to (25), can be represented as

$$D_i F^{ij} = \frac{J^j}{c} \ , \ \ j = \overline{1, 4} \ . \tag{26}$$

Due to the covariance of the Equation (26) with respect to any transformations from the group $GL(4, \mathbb{R})$, they are valid for any continuous currents. These equations are a generalization of the Maxwell equations of four-dimensional electrodynamics in Minkowski space

$$\frac{\partial F^{ij}}{\partial x^i} = \frac{J^j}{c} \ , \ \ j = \overline{1, 4} \ . \tag{27}$$

Before we proceed to analyze the properties of the equations of six-dimensional electrodynamics, it is necessary to make sure that they are closely related to Maxwell's electrodynamic equations.

## 6. The Model of a Resting Electric Charge in Six-Dimensional Electrodynamics and Its Connection with the Analogous Model of Maxwell's Electrodynamics

So far all conclusions have been valid for the case of the presence of both fields: gravitational and electromagnetic. Maxwell electrodynamic equations do not involve gravity. In order to show the connection of six-dimensional theory with traditional four-dimensional Maxwell equations we now resort to the case of absence of gravity.

Geodesics of the form (19) correspond to spaces with symmetric affine connectivity, while geodesics with asymmetric connectivity of the form (20) correspond to spaces with Cartan type torsion. Note that the connectivity (17) and (18) depend on local velocities in addition to their dependence on space-time coordinates. Further we will consider the simpler case of (19).

Equations for geodesics (15) and (16) are valid for spaces of arbitrary dimension and signature. Equation (15) describes geodesics in the presence of gravitational and electromagnetic fields simultaneously. If there is no gravitational field (i.e., $\Gamma^p_{ij} = 0$), the Equation (15) after simple transformations takes the form

$$\ddot{x}^l + F^{pl} \dot{x}_p = 0 \ .$$

The resulting relation is nothing but the Lorentz equation describing the motion of a charged particle with a unit charge in the electromagnetic field defined by the tensor $F^{pl}$. Thus, the vector field $A(x)$, appearing in the definition of the space metric, from a

physical point of view can be interpreted as the field of the vector potential in classical electrodynamics, and the antisymmetric covariant tensor $F_{ik}$ is seen as the electromagnetic tensor. However, if we limit ourselves to considering only the four-dimensional signature space $(1,3)$, the condition (10) arising in the definition of the geodesic does not allow us to build a theory of the interaction of the electromagnetic field with the electric charge (the so-called $A - J$ interaction). In addition, the very notion of electric charge within four-dimensional space-time cannot be given a clear mathematical definition.

To eliminate the difficulties arising in the four-dimensional model of electromagnetism, we have chosen a six-dimensional RWF-space with the signature $(3,3)$ as a candidate for the role of real physical space-time.

In six-dimensional RWF-space on the vector field components $A_k$, $k = 5,6$ in the three-dimensional time subspace we impose some conditions that allow us to derive the system of three-dimensional Maxwell equations. Let us give a geometrical interpretation of the notion of the electric charge distribution density.

In six-dimensional electrodynamics there are two systems of general covariant Equations (21) and (22). The system of Equation (21) consists of twenty relations, the system (22) has six relations. Maxwell's classical electrodynamics in three-dimensional Euclidean space is traditionally represented as two pairs of equations

$$\text{div}E = \rho \ , \quad \text{rot}H - \frac{1}{c}\frac{\partial E}{\partial t} = \frac{j}{c} \ , \tag{28}$$

$$\text{div}H = 0 \ , \quad \text{rot}E - \frac{1}{c}\frac{\partial H}{\partial t} = 0 \ , \tag{29}$$

where $H = (H_1, H_2, H_3)$, $E = (E_1, E_2, E_3)$ are three-dimensional vectors of magnetic and electric field density respectively, $\rho$ is an electric charge density, $j = (j_1, j_2, j_3)$ is a three-dimensional vector of electric charge density.

In order to obtain the system of three-dimensional Maxwell Equations (28) and (29) from the six-dimensional system of Equations (21) and (22) in the case of a resting point electric charge, we should impose some conditions on the covector field $A(x)$. Let $(x_1, \ldots, x_6)$ be the pseudo-Euclidean coordinate system in the tangent layer $T_x$ over the point $x \in M^6$, and the coordinates of the point itself are zero. Hereafter, the coordinates $x_1$, $x_2$, $x_3$ will be called "spatial", and $x_4$, $x_5$, $x_6$ will be referred to as "temporal". In the coordinate plane $(x_5, x_6)$ of the tangent bundle $T_x$ we set a one-parameter rotation group at an angle $\frac{\omega}{c}x^4$ around the origin (coinciding with the tangent point $x$). Then

$$\begin{aligned} x^5(x^4) &= x^5(0)\cos\left(\frac{\omega}{c}x^4\right) - x^6(0)\sin\left(\frac{\omega}{c}x^4\right) \ , \\ x^6(x^4) &= x^5(0)\sin\left(\frac{\omega}{c}x^4\right) + x^6(0)\cos\left(\frac{\omega}{c}x^4\right) \ . \end{aligned} \tag{30}$$

Integral curves $x^5(x^4)$, $x^6(x^4)$, given by the relations (30), generate a vector velocity field, which in the plane $(x^5, x^6)$ is $\left(-\frac{\omega}{c}x^6, \frac{\omega}{c}x^5\right)$. Assume by definition to satisfy Maxwell's equations:

$$(A_5, A_6) = \left(-A\frac{x^6}{\sqrt{x_5^2 + x_6^2}}\delta(x_1, x_2, x_3), A\frac{x^5}{\sqrt{x_5^2 + x_6^2}}\delta(x_1, x_2, x_3)\right) \ , \tag{31}$$

where $A^2 = A_5^2 + A_6^2$, $x_5^2 + x_6^2 = r_0^2 = \text{const}$.

Thus, in the plane of the tangent splitting $T_x$ there is a circulation of the vector field $A$ along the circle of constant radius $r_0$. By virtue of the construction, $A_5$ and $A_6$ are nonzero in the area of definition of the function $\delta(x_1, x_2, x_3) \, \delta(x_5^2 + x_6^2 - r_0^2)$. Regarding the other components of the vector field $A$, suppose by definition that they depend only on the coordinates $x^1, \ldots, x^4$, that is $A_i = A_i(x_1, \ldots, x_4)$, $i = \overline{1,4}$.

Let us introduce the following notations:

$$E_i = F_{i4} = A_{4,i} - A_{i,4} , \quad i = \overline{1,6} ,$$
$$H_1 = F_{23} = A_{3,2} - A_{2,3} , \quad H_2 = F_{31} = A_{1,3} - A_{3,1} ,$$
$$H_3 = F_{12} = A_{2,1} - A_{1,2} , \quad H_4 = F_{56} = A_{6,5} - A_{5,6} .$$

From the definition of the components $A_5$, $A_6$ directly follows that

$$E_5 = -A_{5,4} = \frac{\omega}{c} A \frac{x^5}{\sqrt{x_5^2 + x_6^2}} \delta(x_1, x_2, x_3) ,$$

$$E_6 = -A_{6,4} = \frac{\omega}{c} A \frac{x^6}{\sqrt{x_5^2 + x_6^2}} \delta(x_1, x_2, x_3) ,$$

$$H_4 = \frac{A}{\sqrt{x_5^2 + x_6^2}} \delta(x_1, x_2, x_3)$$

for $x_5^2 + x_6^2 = r_0^2$. The remaining components of the antisymmetric tensor $F_{ij}$ are identically zero. Thus, $F_{ij}$ can be represented in the form

$$F_{ij} = \begin{pmatrix} 0 & H_3 & -H_2 & E_1 & 0 & 0 \\ -H_3 & 0 & H_1 & E_2 & 0 & 0 \\ H_2 & -H_1 & 0 & E_3 & 0 & 0 \\ -E_1 & -E_2 & -E_3 & 0 & -E_5 & -E_6 \\ 0 & 0 & 0 & E_5 & 0 & -H_4 \\ 0 & 0 & 0 & E_6 & H_4 & 0 \end{pmatrix}. \tag{32}$$

Four relations from the system of Equation (21) take the following form for the indices $i, j \in 1, \ldots, 4$:

$$F_{12,3} + F_{23,1} + F_{31,2} = 0 , \quad F_{12,4} + F_{24,1} + F_{41,2} = 0 ,$$
$$F_{13,4} + F_{34,1} + F_{41,3} = 0 , \quad F_{23,4} + F_{34,2} + F_{42,3} = 0 .$$

These relations represent the second pair of Maxwell Equation (29). Apart from (29), the system (21) contains $C_6^3 - 4 = 16$ additional equations, which are identically zero-turned. Let us show this by the example of one of them: $F_{15,2} + F_{21,5} + F_{52,1} = 0$. As the tensor $F_{ij}$ is defined by (32), the components $F_{15}$, $F_{25}$ are identically equal to zero, and $F_{21}$ is by definition independent of $x^5$, hence, the left part of the equation is zero. The same can be shown for the other fifteen equations. Thus, one can conclude that the system of Equation (21) in six-dimensional space-time in the framework of the electromagnetic field model with a two-dimensional vector field component $A(x)$ circulating along the circle in the time subspace is equivalent to the second pair of Maxwell equations.

In order to derive the first pair of Maxwell equations from the system of Equation (22), it is necessary to exclude the effects associated with the presence of the gravitational field and external charges. This is achieved if the pseudo-Riemannian metric, which is a component of the metric of the RWF-space, degenerates into the flat metric of the six-dimensional pseudo-Euclidean signature space $(3,3)$. In this case the antisymmetric tensor $F^{\nu\mu} = g^{i\nu} g^{\mu j} F_{ij}$ takes the form

$$F^{\nu\mu} = \begin{pmatrix} 0 & H_3 & -H_2 & -E_1 & 0 & 0 \\ -H_3 & 0 & H_1 & -E_2 & 0 & 0 \\ H_2 & -H_1 & 0 & -E_3 & 0 & 0 \\ E_1 & E_2 & E_3 & 0 & -E_5 & -E_6 \\ 0 & 0 & 0 & E_5 & 0 & -H_4 \\ 0 & 0 & 0 & E_6 & H_4 & 0 \end{pmatrix}, \tag{33}$$

and the system of Equation (22) degenerates into

$$\frac{\partial F^{\nu\mu}}{\partial x^{\nu}} = 0 , \quad \mu = \overline{1,6} . \tag{34}$$

In the framework of the considered model, with the circulating vector field component $A(x)$ in the time subspace, the system (34) is represented by six equations in the following expanded form:

$$
\begin{aligned}
\frac{\partial F^{\nu 1}}{\partial x^{\nu}} &= -H_{3,2} + H_{2,3} + E_{1,4} = 0 , \\
\frac{\partial F^{\nu 2}}{\partial x^{\nu}} &= H_{3,1} - H_{1,3} + E_{2,4} = 0 , \\
\frac{\partial F^{\nu 3}}{\partial x^{\nu}} &= -H_{2,1} + H_{1,2} + E_{3,4} = 0 , \\
\frac{\partial F^{\nu 4}}{\partial x^{\nu}} &= E_{1,1} + E_{2,2} + E_{3,3} - E_{5,5} - E_{6,6} = 0 , \\
\frac{\partial F^{\nu 5}}{\partial x^{\nu}} &= -E_{5,4} + H_{4,6} = 0 , \\
\frac{\partial F^{\nu 6}}{\partial x^{\nu}} &= -E_{6,4} + H_{4,5} = 0 .
\end{aligned}
\tag{35}
$$

The first three equations of (35) represent the second Maxwell equation in (28) in the absence of currents. The fourth equation from (35) can be written as

$$\mathrm{div} E = E_{5,5} + E_{6,6} , \tag{36}$$

where the right-hand side represents the density of a point electric charge placed at the origin. The Equation (36) is nothing but the first Maxwell equation of the system (28). However, the Equation (36) is more informative because it gives a geometric interpretation of the density of the point electric charge.

Let us now proceed to consider the last two equations of the system (35), which are not part of the system of Maxwell's Equations (28) and (29). These equations are responsible for the relationship between the components of the electromagnetic tensor $F_{ij}$ in time subspace. By virtue of the definition of $E_5$, $E_6$, $H_4$ we have

$$-E_{5,4} + H_{4,6} = \left( \frac{\omega^2 A}{c^2 \left(x_5^2 + x_6^2\right)^{1/2}} x^6 - \frac{A}{\left(x_5^2 + x_6^2\right)^{3/2}} x^6 \right) \delta(x_1, x_2, x_3) ,$$

$$-E_{6,4} + H_{4,5} = \left( \frac{\omega^2 A}{c^2 \left(x_5^2 + x_6^2\right)^{1/2}} x^5 - \frac{A}{\left(x_5^2 + x_6^2\right)^{3/2}} x^5 \right) \delta(x_1, x_2, x_3) ,$$

It follows that the last two equations of the system (35) are equivalent to each other and reduce to a simple algebraic relation

$$\omega^2 r_0^2 = c^2 . \tag{37}$$

From (37) it follows that the linear velocity of the vector field circulation in two-dimensional time subspace is equal to the speed of light. This statement is the basis for understanding why the photons, carriers of electromagnetic field, move in space at the speed of light.

We also suppose that additional temporal dimensions are localized to the scale of elementary particle size, i.e., time span of below $\tau = 10^{-20}$ s. This may explain the difference in conservation laws of strong and weak interactions. Strong interaction has typical time below $\tau$ and occurs in symmetrical $(3,3)$ space, thus it obeys more laws to

keep symmetry. Weak interaction has times above $\tau$ and occurs in less symmetric usual $(1,3)$ space, thus it violates some laws, which are no longer supported by symmetry [17].

### 7. The Derivation of Maxwell's Equations in the Case of Uniformly and Rectilinearly Flowing Currents

The equations of six-dimensional electrodynamics were derived only for the case of a resting electric charge. Now let us derive them for a rather wide class of currents. Let us show that these currents are related to the elements of local groups of proper motions of the Minkowski metric of four-dimensional space and only for this class of currents the equations of electrodynamics adequately describe electromagnetic processes.

In order to obtain Maxwell equations containing currents, let us first consider one particular example of constructing Maxwell equations with uniformly and linearly flowing currents.

To obtain such equations, it is sufficient to move from the coordinate system $(x^1, \ldots, x^6)$, relative to which the charge is at rest, to some other system $(x^{1'}, \ldots, x^{6'})$, relative to which the charge is moving in a straight line with constant speed. This can be done by pseudo-orthogonal rotation. Physically, this means that the transition is made to an inertial frame of reference moving uniformly and linearly relative to the original frame of reference, for example, along the axis $x_1'$. The Lorentz transformations carrying out such transition are

$$x^1 = \gamma\left(x^{1'} + \beta x^{4'}\right), \quad x^2 = x^{2'}, \quad x^3 = x^{3'},$$

$$x^4 = \gamma\left(x^{4'} + \beta x^{1'}\right), \quad x^5 = x^{5'}, \quad x^6 = x^{6'},$$

where $\beta = v/c$ and $\gamma = \left(1 - \beta^2\right)^{-1/2}$ is the relativistic multiplier. The electromagnetic tensor $F_{ij}$, defined by the relation (32), transforms into the tensor

$$F_{i'j'} = \frac{\partial x^i}{\partial x^{i'}} \frac{\partial x^j}{\partial x^{j'}} F_{ij} \tag{38}$$

according to the tensor transformation law. Considering that

$$\frac{\partial x^1}{\partial x^{1'}} = \gamma, \quad \frac{\partial x^1}{\partial x^{4'}} = \gamma\beta, \quad \frac{\partial x^4}{\partial x^{1'}} = \gamma\beta, \quad \frac{\partial x^4}{\partial x^{4'}} = \gamma,$$

we obtain the following representation for the tensor $F_{i'j'}$:

$$F_{i'j'} = \begin{pmatrix} 0 & \gamma(H_3 + \beta E_2) & -\gamma(H_2 - \beta E_3) & E_1 & -\gamma\beta E_5 & -\gamma\beta E_6 \\ -\gamma(H_3 + \beta E_2) & 0 & H_1 & \gamma(E_2 + \beta H_3) & 0 & 0 \\ \gamma(H_2 - \beta E_3) & -H_1 & 0 & \gamma(E_3 - \beta H_2) & 0 & 0 \\ -E_1 & -\gamma(E_2 + \beta H_3) & -\gamma(E_3 - \beta H_2) & 0 & -\gamma E_5 & -\gamma E_6 \\ \gamma\beta E_5 & 0 & 0 & \gamma E_5 & 0 & -H_4 \\ \gamma\beta E_6 & 0 & 0 & \gamma E_6 & H_4 & 0 \end{pmatrix}. \tag{39}$$

As can be seen from (39), the transformed tensor $F_{i'j'}$ contains two additional non-zero components $F_{1'5'}$ and $F_{1'6'}$ as compared to the tensor of the form (32). Let

$$H_{1'} = H_1, \quad H_{2'} = \gamma(H_2 - \beta E_3), \quad H_{3'} = \gamma(H_3 + \beta E_2), \quad H_{4'} = H_4,$$

$$E_{1'} = E_1, \quad E_{2'} = \gamma(E_2 + \beta H_3), \quad E_{3'} = \gamma(E_3 - \beta H_2), \quad E_{5'} = \gamma E_5, \quad E_{6'} = \gamma E_6.$$

Since the pseudo-orthogonal transformation of the pseudo-Euclidean metric remains invariant, we obtain the following representation for the contravariant tensor $F^{i'j'}$:

$$F^{i'j'} = \begin{pmatrix} 0 & H_{3'} & -H_{2'} & -E_{1'} & \beta E_{5'} & \beta E_{6'} \\ -H_{3'} & 0 & H_{1'} & -E_{2'} & 0 & 0 \\ H_{2'} & -H_{1'} & 0 & -E_{3'} & 0 & 0 \\ E_{1'} & E_{2'} & E_{3'} & 0 & -E_{5'} & -E_{6'} \\ -\beta E_{5'} & 0 & 0 & E_{5'} & 0 & -H_{4'} \\ -\beta E_{6'} & 0 & 0 & E_{6'} & -H_{4'} & 0 \end{pmatrix}. \tag{40}$$

It follows that the system of equations $F^{i'j'}_{,i'} = 0$, $j' = 1', 2', 3'$ is equivalent to the equation

$$\text{rot} H' - \frac{1}{c}\frac{\partial E'}{\partial t'} - \beta \rho' = 0, \tag{41}$$

where $H' = (H_{1'}, H_{2'}, H_{3'})$, $E' = (E_{1'}, E_{2'}, E_{3'})$, $\rho' = E_{5'5'} + E_{6',6'} = \gamma(E_{5,5} + E_{6,6}) = \gamma\rho$ is charge density in the primed coordinate system. The value $J' = v\rho'$ is the current density in the primed coordinate system. Then from (41) we have

$$\text{rot} H' = \frac{1}{c}\frac{\partial E'}{\partial t'} + \frac{J'}{c}. \tag{42}$$

The last relation is the Maxwell equation in the case of a uniform and linear current $J'$. The equation $F^{i'4'}_{,i'} = 0$ is written as

$$\text{div} E' = \rho'. \tag{43}$$

There are two more equations $F^{i'5'}_{,i'} = 0$, $F^{i'6'}_{,i'} = 0$, which provide important additional information about the mechanism of electric charge formation, but they are not of direct interest for the purposes stated in this paper. The considered example shows that each proper linear transform from Lorentz group, which is a linear group of eigenmovements of the Minkowski metric, is assigned to a certain linear current from the class of all linearly and uniformly flowing currents. This class of currents is invariant under transformations from the Lorentz group. The question arises whether there are any other groups of coordinate transformations representing groups of eigenmovements of the pseudo-Euclidean metric, acting not on the whole space as it is in the case of the Lorentz group, but only locally on certain classes of trajectories.

## 8. Groups of Local Eigenmovements of the Minkowski Metric and Admissible Classes of Currents

Let a local transformation of coordinate differentials is set in some neighborhood of a fixed point of $n$-dimensional pseudo-Euclidean space with coordinates $x^1, \ldots, x^n$ as:

$$dx^i = \frac{\partial x^i}{\partial x^{j'}} dx^{j'}.$$

Then the pseudo-Euclidean metric $\eta_{ij}$ in the new coordinate system in the neighborhood of a given point takes the form

$$g_{i'j'} = \frac{\partial x^k}{\partial x^{i'}}\frac{\partial x^l}{\partial x^{j'}}\eta_{kl}.$$

**Definition 5.** *The local transformation of coordinate differentials in the neighborhood of a given point that leaves the pseudo-Euclidean metric invariant will be called local eigenmotion of the pseudo-Euclidean metric.*

All possible local transformations of coordinates in the neighborhood of a given point form a group of local eigenmovements of the pseudo-Euclidean metric. Due to the nonlinear nature of the transformations, they may, in general, be non-integrable. Moreover, the question arises about the existence of such groups of nonlinear local motions in principle.

Leaving this question aside for the time being, let us pass to establishment of connection between elements of groups of local motions and admissible classes of currents satisfying Maxwell's equation.

Let a pseudo-Euclidean coordinate system $x^1, \ldots, x^6$ and some local system $x^{1'}, \ldots, x^{6'}$ are set in the neighborhood of some point of six-dimensional pseudo-Euclidean space. Let the nondegenerate Jacobi matrix $|\partial x^i / \partial x^{j'}|$ exist, and let the system $x^1, \ldots, x^6$ contain tensor $F_{ij}$ of form (32). Setting such a tensor (38) in the framework of Maxwell's four-dimensional electrodynamics means that there are no currents in the system. Let's pass to the local primed coordinate system. The components of the electromagnetic tensor in the primed coordinate system take the form

$$
\begin{aligned}
F_{k'l'} =& \left( -H_3 \frac{\partial x^2}{\partial x^{k'}} + H_2 \frac{\partial x^3}{\partial x^{k'}} - E_1 \frac{\partial x^4}{\partial x^{k'}} \right) \frac{\partial x^1}{\partial x^{l'}} \\
&+ \left( H_3 \frac{\partial x^1}{\partial x^{k'}} - H_1 \frac{\partial x^3}{\partial x^{k'}} - E_2 \frac{\partial x^4}{\partial x^{k'}} \right) \frac{\partial x^2}{\partial x^{l'}} \\
&+ \left( -H_2 \frac{\partial x^1}{\partial x^{k'}} + H_1 \frac{\partial x^2}{\partial x^{k'}} - E_3 \frac{\partial x^4}{\partial x^{k'}} \right) \frac{\partial x^3}{\partial x^{l'}} \\
&+ E_i \frac{\partial x^i}{\partial x^{k'}} \frac{\partial x^4}{\partial x^{l'}} \\
&+ \left( H_4 \frac{\partial x^6}{\partial x^{k'}} - E_5 \frac{\partial x^4}{\partial x^{k'}} \right) \frac{\partial x^5}{\partial x^{l'}} \\
&+ \left( -H_4 \frac{\partial x^5}{\partial x^{k'}} - E_6 \frac{\partial x^4}{\partial x^{k'}} \right) \frac{\partial x^6}{\partial x^{l'}} , \quad k', l' = \overline{1', 6'} .
\end{aligned}
\tag{44}
$$

As above, let us introduce the standard notations

$$
H_{1'} = F_{2'3'} , \quad H_{2'} = F_{3'1'} , \quad H_{3'} = F_{1'2'} , \quad H_{4'} = F_{6'5'} , \quad E_{i'} = F_{i'4'} , \quad i' = \overline{1', 6'} .
$$

We require that the local coordinate transformation is a local movement of the pseudo-Euclidean metric of six-dimensional space, or more precisely, it would be an element of the group of local eigenmovements of the metric of the four-dimensional Minkowski subspace. Such a group coincides with the group of local proper transformations of coordinate differentials $x^{1'}, \ldots, x^{4'}$, that is

$$
\begin{aligned}
dx^{j'} &= \frac{\partial x^{j'}}{\partial x^i} dx^i , \quad j' = \overline{1', 4'} , \quad i = \overline{1, 4} , \\
dx^{5'} &= dx^5 , \quad dx^{6'} = dx^6 .
\end{aligned}
\tag{45}
$$

Let us now proceed to the derivation of the first pair of Maxwell Equations (42) and (43). The contravariant tensor $F^{i'j'}$ in the local primed coordinate system will have the form

$$
F_{i'j'} = \begin{pmatrix}
0 & H_{3'} & -H_{2'} & -E_{1'} & E_{5'}\pi_{1'} & E_{6'}\pi_{1'} \\
-H_{3'} & 0 & H_{1'} & -E_{2'} & E_{5'}\pi_{2'} & E_{6'}\pi_{2'} \\
H_{2'} & -H_{1'} & 0 & -E_{3'} & E_{5'}\pi_{3'} & E_{6'}\pi_{3'} \\
E_{1'} & E_{2'} & E_{3'} & 0 & -E_{5'} & -E_{6'} \\
-E_{5'}\pi_{1'} & -E_{5'}\pi_{2'} & -E_{5'}\pi_{3'} & E_{5'} & 0 & -H_{4'} \\
-E_{6'}\pi_{1'} & -E_{6'}\pi_{2'} & -E_{6'}\pi_{3'} & E_{6'} & H_{4'} & 0
\end{pmatrix} ,
\tag{46}
$$

where $H_{4'} = H_4$ and

$$
\pi_{m'} = \frac{\partial x^4}{\partial x^{m'}} \left( \frac{\partial x^4}{\partial x^{4'}} \right)^{-1} .
$$

The system of equations $F^{i'j'}_{,i'} = 0$, $j' = \overline{1', 6'}$ in the primed local coordinate system can be represented in the form

$$\mathrm{rot}H' - \frac{1}{c}\frac{\partial E'}{\partial t'} - \frac{J'}{c} = 0 \,,$$

$$\mathrm{div}E' = \rho' \,,$$

$$\frac{\partial}{\partial x^{1'}}(E_{5'}\pi_{1'}) + \frac{\partial}{\partial x^{2'}}(E_{5'}\pi_{2'}) + \frac{\partial}{\partial x^{3'}}(E_{5'}\pi_{3'}) - \frac{\partial E_{5'}}{\partial x^{4'}} + \frac{\partial H_{4'}}{\partial x^{6'}} = 0 \,,$$

$$\frac{\partial}{\partial x^{1'}}(E_{6'}\pi_{1'}) + \frac{\partial}{\partial x^{2'}}(E_{6'}\pi_{2'}) + \frac{\partial}{\partial x^{3'}}(E_{6'}\pi_{3'}) - \frac{\partial E_{6'}}{\partial x^{4'}} - \frac{\partial H_{4'}}{\partial x^{5'}} = 0 \,,$$

(47)

where $\rho' = E_{5',5'} + E_{6',6'}$, $J' = \rho' v$, $v/c = (-\pi_{1'}, -\pi_{2'}, -\pi_{3'})$.

The last two equations of the system (47) are equivalent to the following

$$\frac{\partial}{\partial x^{1'}}\left(E_5\frac{\partial x^4}{\partial x^{1'}}\right) + \frac{\partial}{\partial x^{2'}}\left(E_5\frac{\partial x^4}{\partial x^{2'}}\right) + \frac{\partial}{\partial x^{3'}}\left(E_5\frac{\partial x^4}{\partial x^{3'}}\right) - \frac{\partial}{\partial x^{4'}}\left(E_5\frac{\partial x^4}{\partial x^{4'}}\right) + \frac{\partial H_{4'}}{\partial x^{6'}} = 0 \,,$$

$$\frac{\partial}{\partial x^{1'}}\left(E_6\frac{\partial x^4}{\partial x^{1'}}\right) + \frac{\partial}{\partial x^{2'}}\left(E_6\frac{\partial x^4}{\partial x^{2'}}\right) + \frac{\partial}{\partial x^{3'}}\left(E_6\frac{\partial x^4}{\partial x^{3'}}\right) - \frac{\partial}{\partial x^{4'}}\left(E_6\frac{\partial x^4}{\partial x^{4'}}\right) - \frac{\partial H_{4'}}{\partial x^{5'}} = 0 \,.$$

(48)

Taking into account that

$$\frac{\partial}{\partial x^{k'}} = \frac{\partial x^i}{\partial x^{k'}}\frac{\partial}{\partial x^i} \,,$$

and also the independence of the components $E_5$, $E_6$ from the variables $x^1$, $x^2$, $x^3$, we can transform the system (48) to the form

$$D\frac{\partial E_5}{\partial x^4} + \frac{\partial H_4}{\partial x^6} = 0 \,,$$

$$D\frac{\partial E_6}{\partial x^4} - \frac{\partial H_4}{\partial x^5} = 0 \,,$$

$$D = \left(\frac{\partial x^4}{\partial x^{1'}}\right)^2 + \left(\frac{\partial x^4}{\partial x^{2'}}\right)^2 + \left(\frac{\partial x^4}{\partial x^{3'}}\right)^2 - \left(\frac{\partial x^4}{\partial x^{4'}}\right)^2 \,.$$

(49)

Since the transition from the coordinate system $x^1, \ldots, x^4$ to the system $x^{1'}, \ldots, x^{4'}$ is performed using a local pseudo-orthogonal transformation, the condition $D = 1$ must be satisfied. It follows that the system (49) is equivalent to the system

$$-\frac{\partial E_5}{\partial x^4} + \frac{\partial H_4}{\partial x^6} = 0 \,, \quad \frac{\partial E_6}{\partial x^4} + \frac{\partial H_4}{\partial x^5} = 0 \,. \tag{50}$$

These two equations provide important additional information about the relationship between the components of the electromagnetic tensor $F^{i'j'}$, but they are not of direct interest for the purposes stated in this paper, and therefore will not be considered here.

Thus, the system of Equation (47) of six-dimensional electrodynamics in the considered model is invariant under any local pseudo-orthogonal coordinate transformations $x^1, \ldots, x^4$ and equivalent to the first pair of Maxwell equations and additional system of two Equation (50). Note that the three-dimensional current vector $J'$ included in the first Maxwell equation of the system (47) is completely determined by a kind of local pseudo-orthogonal transformation, according to the formula

$$\frac{J'}{c} = \rho'(\pi_{1'}, \pi_{2'}, \pi_{3'}) = \rho'\left(\frac{\partial x^4}{\partial x^{1'}}\left(\frac{\partial x^4}{\partial x^{4'}}\right)^{-1}, \frac{\partial x^4}{\partial x^{2'}}\left(\frac{\partial x^4}{\partial x^{4'}}\right)^{-1}, \frac{\partial x^4}{\partial x^{3'}}\left(\frac{\partial x^4}{\partial x^{4'}}\right)^{-1}\right) \,. \tag{51}$$

Thus, each admissible current included in the first Equation (47) is defined by some element of the group of local eigenmovements of the Minkowski metric according to (51). All of the above can be formulated as the following result.

**Theorem 1.** *Maxwell's equations are invariant with respect to any local eigenmovements of the Minkowski metric. The class of admissible currents satisfying Maxwell's equations is completely determined by the group of local eigenmovements of the Minkowski metric.*

It follows from the theorem that the wider the group of local eigenmovements of the Minkowski metric, the larger the class of admissible currents satisfying Maxwell's equation, and the wider is the field of application of Maxwell's electrodynamics. Indeed, if the group of local eigenmovements were limited to Lorentz transformations only, then Maxwell's equations would be strictly valid only in the case of uniformly and linearly flowing currents. This fact was pointed out at the time by Pauli [18]. The question about the existence of groups of local eigenmovements of the Minkowski metric, different from the Lorentz group, remains open.

### 9. An Example of a Nonlinear Group of Local Eigenmovements of a Minkowski Metric

An arbitrary eigenrotation of three-dimensional pseudo-Euclidean space $\mathbb{R}^3_{1,2}$, that is an arbitrary orthogonal or pseudo-orthogonal transformation leaving the origin of coordinates immobile, can be decomposed into three rotations in the planes $\{x^1, x^2\}$, $\{x^1, x^4\}$, $\{x^2, x^4\}$ and one rotation in space itself $\mathbb{R}^3_{1,2} = \{x^1, x^2, x^3\}$, which is not reduced to any of the previous ones. The first rotation transforms the spatial coordinates $x^1$, $x^2$ and corresponds to the usual spatial rotations. The second and third rotations act in pseudo-Euclidean planes and correspond to the Lorentz proper transforms. Let us pass to consideration of the fourth rotation in $\mathbb{R}^3_{1,2}$. The sought rotation should leave invariant the differential quadratic form

$$\left(dx^4\right)^2 - \left(dx^1\right)^2 - \left(dx^2\right)^2 \tag{52}$$

or an equivalent form written in polar coordinates

$$\left(dx^4\right)^2 - dr^2 - r^2 d\varphi^2 , \tag{53}$$

where $x^1 = r \cos\varphi$, $x^2 = r \sin\varphi$.

Let the coordinate differentials $x^4$, $\varphi$ of a point with coordinates $(x^4, r, \varphi)$ undergo a transformation $a_\omega : (dx^4, d\varphi) \to (dx^{4'}, d\varphi')$ of the form

$$dx^4 = \frac{dx^{4'} + \frac{r^2\omega}{c} d\varphi'}{\sqrt{1 - \left(\frac{r\omega}{c}\right)^2}} , \quad d\varphi = \frac{d\varphi' + \frac{\omega}{c} dx^{4'}}{\sqrt{1 - \left(\frac{r\omega}{c}\right)^2}} , \tag{54}$$

where $\omega$ is the angular velocity of rotation of a circle of radius $r$ in the plane $\{x^1, x^2\}$ with respect to the origin, $|\omega| = x/r$. It is easy to see that the coordinate transformation (54), which leaves the invariant form (53), is an element of the sought group of local eigenmovements of the Minkowski metric. If we return to the pseudo-Euclidean coordinate system $(x^4, x^1, x^2)$, the transformations (54) will be equivalent to the following nonlinear transformations

$$dx^4 = \gamma dx^{4'} - \gamma \frac{\omega}{c} x^{2'} dx^{1'} + \gamma \frac{\omega}{c} x^{1'} dx^{2'} ,$$

$$dx^1 = -\gamma \beta \sin \varphi dx^{4'} + \frac{\omega}{v} A dx^{1'} + \frac{\omega}{v} B dx^{2'} ,$$

$$dx^2 = \gamma \beta \cos \varphi dx^{4'} + \frac{\omega}{v} C dx^{1'} + \frac{\omega}{v} D dx^{2'} ,$$

$$\varphi = \gamma \left( \frac{\arccos x^{1'}}{\sqrt{(x^{1'})^2 + (x^{2'})^2}} + \frac{\omega}{c} x^{4'} \right) , \quad v = r\omega , \tag{55}$$

$$A = x^{1'} \cos \varphi + \gamma x^{2'} \sin \varphi , \quad B = x^{2'} \cos \varphi - \gamma x^{1'} \sin \varphi ,$$

$$C = x^{1'} \sin \varphi + \gamma x^{2'} \cos \varphi , \quad D = x^{2'} \sin \varphi - \gamma x^{1'} \cos \varphi .$$

The transformations (55) leave the differential quadratic form (52) invariant and therefore are local eigenmovements of the Minkowski metric.

## 10. Conclusions

A space with a metric tensor of a special kind depending on the coordinates and local velocities (RWF-space) is introduced. Setting the tensor induces some corresponding field in this space. Geodesic lines are defined by second-order differential equations, the coefficients in which can be divided into those depending on the metric tensor (relating to the gravitational interaction) and those depending on the vector field (relating to the electromagnetic interaction). If there is no gravity, the geodesic equations turn into the equations describing the charge motion in the electromagnetic field.

For the RWF-space with symmetric signature $(3, 3)$ and without gravity a six-dimensional model of electrodynamics is constructed and Maxwell's electrodynamics equations for a certain class of currents are derived. Within the framework of this model, a purely geometrical interpretation of the concept of electromagnetic field and point electric charge was proposed. The first concept owes its emergence to the type of RWF-space metric. The appearance of the point electric charge is associated with the circulation of the vector potential around a dedicated time axis in the three-dimensional time subspace. Thus, the electric charge formation occurs in the unobservable three-dimensional temporal region of six-dimensional space-time, and its existence is manifested in those effects which are observed in the real three-dimensional physical subspace. Additional temporal dimensions in the considered model turn out to be compactified. Because of symmetry in six-dimensional electrodynamics, it would be more correct to abandon the concept of electric charge and operate only with components of the electromagnetic tensor $F_{ij}$ in six-dimensional space-time. Traditionally, Maxwell's equations in the four-dimensional theory of electromagnetism are interpreted as relationships between the spatial distribution of the charge density and the electromagnetic field density, i.e., between phenomenological objects without a clear mathematical definition. In six-dimensional electrodynamics, this interpretation changes to a more rigorous and consistent understanding of the equations of electromagnetism as equations linking the different components of the electromagnetic tensor $F_{ij}$ in six-dimensional space.

The proposed six-dimensional model of classical electrodynamics, apparently, may help to take a new look at the renormalization problem in quantum electrodynamics. As is known [19], the permutation functions and Green's functions have singularities on the light cone of four-dimensional space-time. In six-dimensional electrodynamics due to taking into account the mechanism of electric charge formation the light cone is replaced by a one-band hyperboloid, which should lead to a revision of the calculation technique. Probably, occurrence of meaningless expressions at calculations within traditional four-dimensional quantum electrodynamics is connected exactly with wrong choice of dimensionality and structure of real physical space-time.

It is shown that the Maxwell equations are invariant with respect to the group of local eigenmovements of the Minkowski metric, which is wider than the Lorentz group.

A mutually unambiguous relation has been established between the admissible currents included in the Maxwell equations and the local eigenmovements of the Minkowski metric. An attempt to extend these results to arbitrary currents leads to a change in the form of Maxwell's equations. This means that the Maxwell equations turn out to be valid not for arbitrary currents, as it is accepted at present, but only for a certain class of currents defined by the maximum local group of eigenmovements of the Minkowski metric.

**Author Contributions:** Conceptualization, N.P.; Formal analysis, N.P. and I.M.; Writing—original draft, N.P. and I.M.; Writing—review and editing, I.M. All authors have read and agreed to the published version of the manuscript.

**Funding:** This research received no external funding.

**Institutional Review Board Statement:** Not applicable.

**Informed Consent Statement:** Not applicable.

**Data Availability Statement:** Not applicable.

**Conflicts of Interest:** The authors declare no conflict of interest.

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
