# Peer review of "Six-Dimensional Manifold with Symmetric Signature in a Unified Theory of Gravity and Electromagnetism"

_symmetry, doi:10.3390/sym14061163_

Round 1

Reviewer 1 Report

In this paper, as a spatial structure for developing a combined theory of gravity and electromagnetic, a six-dimensional manifold with symmetric signature (3, 3) is proposed. The space that combines the qualities of Riemann, Weyl, and Finsler spaces is presented using a special metric tensor. The coefficients of geodesic line equations can be classified into two categories: depending on the metric tensor (related to gravitational interaction) and dependent on the vector field (relating to the electromagnetic interaction).
The paper is interesting and I recommend publishing it after a minor revision.
-  References should be updated and some recent references added.

Author Response

Dear Reviewer,

Thank you very much for your assistance. According to your request we have added some recent publications on the topic. Changes are outlined with font color in the updated text, please see the attachment.

Reviewer 2 Report

In the paper entitled” Six-dimensional manifold with symmetric signature in a unified theory of gravity and electromagnetism” a space combining elements of the structures of Riemann, Weyl, and Finsler spaces is introduced for a uniform description of gravitational and electromagnetic interactions. This work has been competently carried out. The manuscript could be surely published in the Symmetry journal as most parts of the observations are reasonably correct and well discussed; I recommend the paper be accepted for publication as is.  

Author Response

Dear Reviewer,

Thank you very much for your assistance. We have revised the paper according to remarks. Changes are outlined with font color in the updated text, please see the attachment.

Reviewer 3 Report

In this paper, the authors study the space with a metric tensor of a special kind depending on the coordinates
and local velocities. For the six dimensional RVF-space and without gravity, a model of electrodynamics is constructed and Maxwell’s electrodynamics
equations for a certain class of currents are derived. The idea is interesting and the paper is written in a highly technically way. Therefore, it deserves to be published.  But before publication, several important questions should be clarified.

1. Why they unify the gravity and electromagnetism in the SIX dimensional spacetime? or, why SIX dimensions is special. What is the size of extra two dimensions? Are the size or effects of extra dimensions consistent with the gravity experiments and astronomical observations?

2. The four dimensional Maxwell equations are derived in the EVF space. However, as the unification of gravity and electromagnetic field, the equation of motion for gravity is not given. In particular, is there any differences between the resulting gravity equations and Einstein equations?

In all, I think these questions are important and must be clarified before  publication. 

Author Response

Dear Reviewer,

Thank you very much for your work with our paper. Considering your remarks:

  1. Six dimension space is chosen as the minimum one, allowing to construct a unified gravitational and electromagnetic theory. Of course, next symmetric-signature spaces, like (4, 4) can be used, but this is unnecessary redundant. The explanation is added at pages 2 and 3, outlined by red font color. We suppose the size of additional two dimensions is in particle scale. The explanation is added at page 15. It is hard to judge now how this can reflect in astronomical scale, we continue the research.
  2. This paper is introduces the mechanism of unification and delves a bit into electromagnetism, leaving gravity apart. We plan to explain gravity aspect in the next publication.Explanation is added at page 8. 

Round 2

Reviewer 3 Report

In the revision, the manuscript has bee improved. It is now ready for
publication.